 SHORT REPORT

# Female-dominated disciplines have lower evaluated research quality and funding success rates, for men and women

Alex James[1]*, Franca Buelow[2,3], Liam Gibson[1], Ann Brower[3]

[1]School of Mathematics and Statistics, University of Canterbury, Christchurch, New Zealand; [2]Bioprotection Centre of Research Excellence, University of Lincoln, Lincoln, Lincoln, New Zealand; [3]School of Earth and Environment, University of Canterbury, Christchurch, New Zealand

**Abstract** We use data from 30 countries and find that the more women in a discipline, the lower quality the research in that discipline is evaluated to be and the lower the funding success rate is. This affects men and women, and is robust to age, number of research outputs, and bibliometric measures where such data are available. Our work builds on others' findings that women's work is valued less, regardless of who performs that work.

## eLife assessment

This study provides **convincing** evidence that the quality of research in female-dominated fields of research is systematically undervalued by the research community. The authors' findings are based on analyses of data from a research assessment exercise in New Zealand and data on funding success rates in Australia, Canada, the European Union and the United Kingdom. This work is an **important** contribution to the discourse on gender biases in academia, underlining the pervasive influence of gender on whole fields of research, as well as on individual researchers.

## Introduction

Many have observed that women's research is evaluated as lower quality than men's research (*Brower and James, 2020*; *West et al., 2013*). And apart from a few exceptions (*Ceci et al., 2023*; *Ceci et al., 2014*; *Albers, 2015*), most analyses show that women's funding success rates are lower than men's (*Witteman et al., 2019*; *Grogan, 2019*; *Brower and James, 2023*) both in terms of number of grants and size of grants (*Grogan, 2019*; *Ley and Hamilton, 2008*). Evaluations of research quality affect many aspects of an academic career, from hiring and promotion decisions to funding success. Other studies have explored possible explanations for this seemingly gendered pattern in evaluation of research quality, such as societal (*Brower and James, 2020*; *Brower and James, 2023*; *Hunter and Leahey, 2010*), institutional (*James and Brower, 2022*), and academic factors (*El-Alayli et al., 2018*; *O'Meara et al., 2017*), including gender bias (*Shen, 2013*).

We find a strong correlation between the evaluated quality of a researcher's work and the gender balance of the researcher's discipline. Researchers, both male and female, in female-dominated disciplines receive lower evaluations. This finding is robust to bibliometric differences across disciplines. We also find that funding success rates display a similar correlation. Applications from researchers, both male and female, in female-dominated disciplines have lower success rates.

Academic disciplines can be quite different to one another in ways such as publication rates (*Olejniczak et al., 2022*), citation rates (*Lancho-Barrantes et al., 2010*), and expected h-indices

*For correspondence:
alex.james@canterbury.ac.nz

**Competing interest:** The authors declare that no competing interests exist.

**eLife digest** There have been growing concerns around sexism in science. Studies have found that women in science are often paid less, are less likely to get credit for their work and receive fewer and smaller grants than men at similar stages in their careers. This can make it harder for women to advance in their careers, resulting in less women than men taking up positions of leadership.

There are also gender imbalances between scientific disciplines, with a higher proportion of women working in some fields compared to others. Here, James et al. set out to find whether having more women working in a discipline leads to biases in how the research is evaluated.

The team examined four datasets which included information on the research evaluations and funding success of thousands of researchers across 30 different countries. The analysis suggested that scientists working in women-dominated disciplines were less likely to succeed in their grant applications. Their research was also often evaluated as being lower quality compared to researchers working in fields dominated by men. These biases applied to both men and women working in these disciplines. There were not sufficient data to analyse patterns faced by non-binary individuals.

The study by James et al. cannot pinpoint a specific cause for these outcomes. However, it suggests that funding organisations should analyse the pattern of successful applications across disciplines and consider taking steps to ensure all disciplines have similar success rates. James et al. also propose that when hiring or making promotions, scientific institutions should take care when comparing researchers across disciplines and ensure there is no built-in assumption that fields dominated by men are intrinsically better.

(*Harzing, 2008*). There are different norms for the expected number of co-authors (*Puuska et al., 2014*), order of authors, and attributions (*Street et al., 2010*). Gender balance is not constant across disciplines either. There are fewer women in STEM subjects, and more women than men use qualitative methods, rather than quantitative, in research (*Shen, 2013*). Recent research suggests there are interactions between individual- and discipline-level characteristics. Higher 'expectations of brilliance' in a discipline is related to a lower proportion of women entering *Leslie et al., 2015*; and research area is a strong contributor to the lower funding success rates of African American/Black scientists (*Hoppe et al., 2019*).

That women score lower on evaluations of research quality is not surprising. On an individual level, women publish less than men (*West et al., 2013*; *Mairesse and Pezzoni, 2015*; *Symonds et al., 2006*), receive fewer citations (*Dion et al., 2018*; *Larivière et al., 2013*), and have lower h-indices (*Chatterjee and Werner, 2021*). Women have fewer first authorships (*West et al., 2013*) and are less likely to receive acknowledgement (*Ross et al., 2022*). Women are also less likely to get credit for their innovations (*Hofstra et al., 2020*; *Ding et al., 2006*) and awards, particularly prestigious ones, are less likely to be given to women (*James et al., 2019*; *Meho, 2021*). Recent research suggests women react differently than men to a change in a journal's prestige or impact factor (*Schmal et al., 2023*).

However, our research shows a different correlation: a higher proportion of women in a discipline is correlated with lower research quality evaluations and funding success rates, for everyone in the field.

We use independent datasets spanning thousands of researchers and 30 countries and look separately at research funding success rates and holistic research quality scores. Where possible we account for researcher characteristics like age, research institution, and publishing patterns. We consistently find that researchers in male-dominated disciplines have both higher funding success rates, and higher research quality scores, than researchers in female-dominated disciplines. This applies to both men and women.

## Methods

We analysed research quality and funding success separately. We used data from four sources (Performance-Based Research Fund [PBRF], Australian Research Council [ARC], Canadian Institute of Health Research [CIHR], European Institute of Gender Equality [EIGE]), spanning 30 countries (described fully in Supplementary Materials). The ARC, CIHR, and EIGE datasets cover funding

**Table 1.** A summary of the four independent datasets used in the study showing: country, time period of the data, number of disciplines, research evaluation method, number of individuals or applicants by gender, number of individual data points, output variable, and mean expected output for each gender across the whole dataset.

| Scheme | PBRF | | | ARC | CIHR | EIGE |
|---|---|---|---|---|---|---|
| Country | Aotearoa New Zealand | | | Australia | Canada | Predominantly EU |
| Time span | 2000–06 | 2007–12 | 2013–18 | 2010–19 | 2011–16 | 2019 |
| Disciplines | 42 | 42 | 43 | 22 | 4 | 8 |
| Evaluation | Research quality score (200–700) | | | Funding success | | |
| Women | 1708 | 2658 | 3297 | 46,231 | 8143 | 49,863 |
| Men | 2522 | 4005 | 4181 | 130,331 | 15,775 | 85,305 |
| Data points | 4230 | 6663 | 7487 | 440 | 16 | 333 |
| Outcome | Mean score | | | Success rate | | |
| Women | 272 | 368 | 379 | 25.7% | 14.7% | 27.0% |
| Men | 351 | 424 | 433 | 27.1% | 16.4% | 30.3% |

success rates across 29 countries. The PBRF dataset covers research quality evaluations for all university academic staff in New Zealand. Summary data from the four datasets are reported in *Table 1*. The discipline category definitions varied in each dataset, e.g., EIGE data used Science as a discipline whereas PBRF broke this down into smaller groupings like physics, biology, etc. Supplementary Materials contains full information for each dataset. Due to data limitations, we consider gender as binary, data on non-binary individuals were either not available (EIGE and CIHR) or the group was too small to be included (PBRF and ARC).

*Research quality evaluations*: We use the Aotearoa New Zealand (ANZ) PBRF data. This dataset contains holistic research portfolio evaluations of every research-active academic in ANZ over three time spans (2000–06, 2007–12, 2013–18). The holistic research portfolio includes an individual's self-nominated best three or four publications, in full, with descriptions of the importance and impact of the publications. It also includes non-publication aspects such as contributions to the research environment like chairing a conference organisation committee. Evaluation is done by subject area panels of experts, who read the publications and evaluate the individual's holistic contribution. It is not an automated process and does not explicitly use measures such as h-indices, number of publications, or number of citations. The evaluation explicitly claims to focus on quality over quantity (*Commission TE, 2017*). Panels are likely to be more gender balanced than the fields they assess but in the more heavily skewed fields (e.g. Education, Engineering, Mathematics) they are skewed as would be expected. The PBRF is described in more detail in Supplementary Materials and summary statistics by year and discipline are given in *Supplementary file 2*.

With over 18,000 individual data points this dataset is a unique opportunity to explore a question such as this. To examine the robustness of any findings to bibliometric measures, we use a subset of

**Table 2.** Maximal candidate model and best fit model to predict *Score* for Performance-Based Research Fund (PBRF) and $logit\left(P\left(Success\right)\right)$ for each dataset.
Wilkinson notation is used to indicate interaction terms.

| | Response | Maximal model | Best model (lowest AIC) |
|---|---|---|---|
| PBRF | *Score* | $Age + \left(1|Inst\right) + f\left(Gender, p_{men}\right)$ | $Age + \left(1|Inst\right) + Gender * p_{men}$ |
| ARC | | $f\left(Gender, p_{men}, Time\right)$ | $\left(Gender + p_{men}\right) * Time$ |
| CIHR | $logit\left(P\left(\begin{smallmatrix} grant \\ success \end{smallmatrix}\right)\right)$ | $f\left(Gender, p_{men}, Time\right)$ | $Gender + p_{men} + Time$ |
| EIGE | | $\left(1|Country\right) + f\left(Gender, p_{men}\right)$ | $\left(1|Country\right) + Gender * p_{men}$ |

the PBRF data from one university to compare anonymised detailed research output data to each individual's PBRF score.

*Funding success rates*: We use three independent publicly available datasets: ARC application success rates from 2010 to 2019; Canadian Institute of Health Research application data from 2013 to 2016; and data covering government research grants across the EU and UK in 2019 from the EIGE. These datasets all provided aggregated data on the number of applications and successes by gender for each research discipline, i.e., not individual-level data.

The datasets are briefly described in the text, *Table 1*, and *Supplementary file 1*, and full data details are in Supplementary Materials. Each dataset is analysed separately and the analysis includes the sample size of each aggregated data point. We define a maximal model including two-way interactions between the three key variables: the gender balance of the discipline measured by the proportion of researchers who are men, $p_{men}$; researcher gender, *Gender*; and, where appropriate, *Time*. Any other available variables, e.g., researcher age, institute, or country were included as linear terms with either a fixed (Age) or random (Institute or Country) effect. We measure 'best' by the lowest AIC and the choice is robust by other similar measures, e.g., maximising Pearson's r-squared or omitting non-significant variables. *Table 2* shows the maximal models and the best models for each of the datasets.

## Scoring research quality

First, we look at the only country that scores all individual university researchers. The ANZ PBRF scores are individual-level micro-data on the results of 18,371 holistic research quality assessment scores of 13,555 individuals in ANZ for the PBRF assessments. The assessments were carried out at three separate time points. In each of 2006, 2012, and 2018, staff members at all of the eight universities in ANZ were given a research performance score from 0 to 700 based on a holistic assessment of their research portfolio over the previous 6 years. Individuals with scores over 200 were considered research active. At each assessment, between 4000 and 8000 researchers were assessed. The scores, based on assessments by local and international experts on 14 different subject area panels, were used to allocate government performance-based research funding to universities. Although many researchers were present for more than one round, we analyse each of the three time points separately to examine the link between research score and the gender balance of a researcher's discipline.

We use a linear regression model to predict the research quality score using the available variables, in particular the gender balance of the discipline, $p_{men}$. Although the data is strictly count data requiring a Poisson model we use a simple linear model as all the predicted values are large, i.e., between 200 and 700. We start with the maximal model of *Table 2* that includes our key variables of interest *Gender* and $p_{men}$ with interactions and the confounding variables *Age* and *Institute* as linear terms with no interactions. We compare all candidate models up to the maximal model and choose

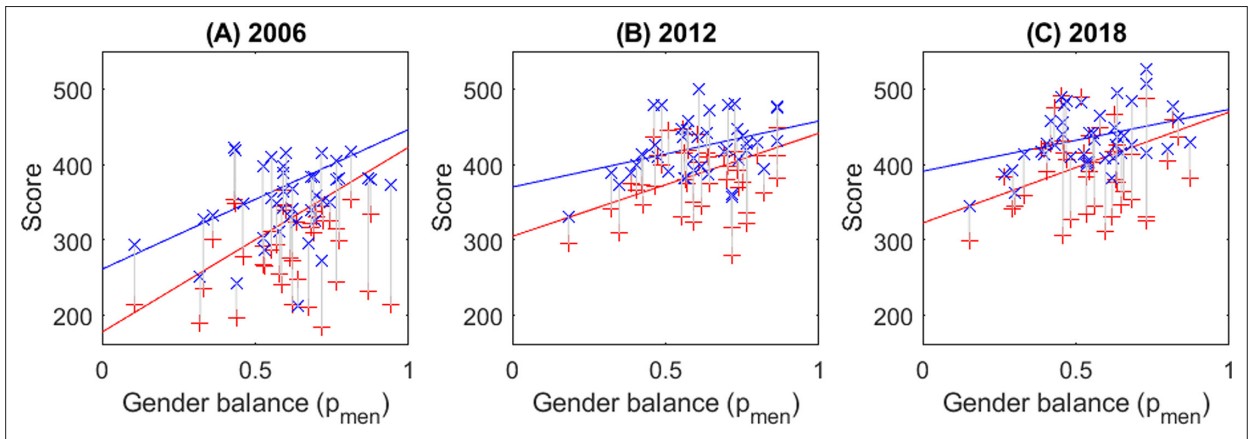

**Figure 1.** Male-dominated disciplines have higher expected research scores than female-dominated disciplines in all three Performance-Based Research Fund (PBRF) assessments. Points – the mean raw score of individuals in each of the disciplines (full raw data cannot be shown for privacy reasons), against the proportion of men in the discipline (blue – men, red – women). Lines – expected score from individual-level analysis ($N = 4135, 6586, 7467$ respectively) adjusting for age, institute, gender, and proportion of men in the discipline, and shown for a 50-year-old man (blue) and woman (red) at one university in Aotearoa NZ. (**A**) 2006 assessment round, (**B**) 2012 assessment round, (**C**) 2018 assessment round.

the best model by minimising AIC. The *Supplementary file 3* gives the full output from all candidate models. For all datasets all terms in the best model (PBRF3 in *Supplementary file 3*) were significant (p < 0.05) and the best model had a $\Delta AIC > 2$ compared to the second best.

The best model for all three time points was

$$Score \sim Age + Institute + Gender * p_{men}.$$

The model predicts that individuals in disciplines dominated by men have higher research performance scores than individuals in female-dominated disciplines; all variables are significant (p < 0.05, see *Supplementary file 3*) at all time points.

Even after accounting for age in this way, men have higher scores than women; but the difference between the most male-dominated disciplines and the most female-dominated is larger than the gap between men and women within any particular discipline. *Figure 1* shows the mean raw score of each discipline by gender (women red +, men blue x) in each assessment period (*Figure 1A*) 2006, (*Figure 1B*) 2012, (*Figure 1C*) 2018. The line is the expected model output for a 50-year-old male (blue line) or female (red line) researcher at the University of Canterbury (UC) (other ages and institutions differ by a constant as they are included as linear terms with no interactions). Note that the full model predictions (lines on *Figure 1*) are adjusted for age so may appear different from the raw data (points on *Figure 1*). In particular, raw data points from discipline/gender groups with many young researchers will appear much lower than might be expected at first thought.

By way of example, we consider a 50-year-old researcher in 2018 at UC (i.e. lines in *Figure 1C*), *Institute* and *Age* are linear terms in the model so a different aged researcher at another university would show a very similar pattern. We assume our researcher comes from either a male-dominated discipline ($p_{men} = 75\%$) like Philosophy or Physics; a gender-balanced discipline ($p_{men} = 50\%$) such as Psychology or Law; or a female-dominated discipline ($p_{men} = 25\%$) like Nursing or Education. In the gender-balanced discipline (Psychology or Law) a woman's expected score is 396 out of a possible 700. This is around 8% lower than a man's expected score of 432/700 in the same discipline. If a man works in a male-dominated discipline (Physics or Philosophy), his mean score is now expected to be higher at 452/700. If a man works in a female-dominated discipline (Nursing or Education), his score will drop to 411/700. In comparison, a female Philosopher or Physicist can expect a score of 433/700, only slightly lower than a male researcher in the same discipline. If a woman works in Nursing or Education, her expected score is now 359/700, a drop of over 20% from the female Philosopher's expected score and also lower than the male Nursing researcher's score.

This effect, of researchers in female-dominated disciplines being predicted to have lower scores than researchers in male-dominated disciplines, persists for all three assessment rounds but was much larger in the 2006 round (*Figure 1A*). In all three rounds, the score drop associated with moving from a male- to a female-dominated discipline is larger for women than for men. Finally, the gender gap within a discipline is much larger in female-dominated disciplines than in male-dominated ones, though the gap has closed considerably between 2006 and 2018.

## Accounting for bibliometric differences between individuals

Here, we explore whether gendered patterns in publication explain the variation in research scores across disciplines. We show that publications are important but they do not negate our previous result that researchers in female-dominated disciplines receive lower research scores.

We analysed a subset of data available from UC, one of the eight universities contained in the PBRF datasets. We included research-active staff (i.e. PBRF score >200) with at least one publication between 2013 and 2018, working at UC in 2018. Using date of birth, gender, ethnicity, and discipline we were able to link the university's internal research publications dataset to individual PBRF scores for over 80% of individuals ($N = 384$, 141 female, 243 male). Data records that could not be linked were excluded.

These publication records allowed us to calculate a series of research bibliometrics, e.g., number of outputs, number of citations, expected field weighted citation index of publications, etc., for each researcher. We then used these bibliometrics to predict each individual's expected research score (see Supplementary Materials for a more detailed description of the data and the analysis). The bibliometrics were tested separately in an individual regression model

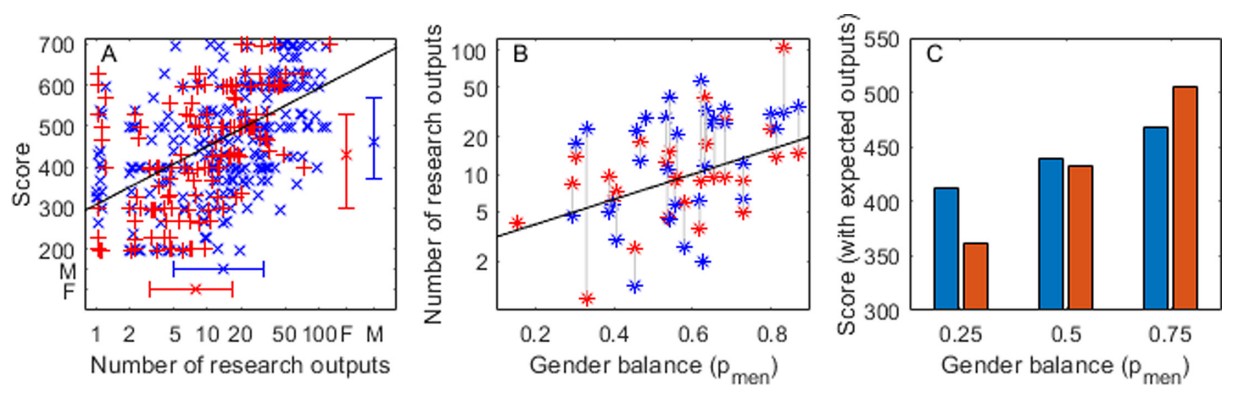

**Figure 2.** Research score is strongly correlated with number of research outputs (A), which is correlated to the gender balance of the discipline (B); but, after adjusting for a typical number of research outputs, female-dominated disciplines still have lower scores (C). Performance-Based Research Fund (PBRF) data using individuals at University of Canterbury (UC) (N=384). (**A**) Score against number of research outputs over the period of the PBRF assessment. Trend line shows the expected score for a 50-year-old. Error bars show the median and interquartile range for men and women for score (vertical) and number of outputs (horizontal). (**B**) Mean number of outputs by gender for each discipline against gender balance of the discipline. Trend line from individual-level analysis ($N = 384$), gender was not significant. (**C**) Expected score of a 50-year-old man and woman with the expected number of outputs for a discipline of that gender balance (25%, 50%, 75% men, respectively).

$$Score \sim Age + Gender + log\left(Bibliometric\right).$$

The bibliometric that was the best predictor of research score (with minimum AIC and maximum r-squared) was the total number of outputs $N_{outputs}$ (see **Supplementary file 4**, model 2). This output gave a model correlation of $r^2 = 0.325$ and the coefficient of $N_{outputs}$ was highly significant ($p < 0.001$). In this model *Age* was also significant ($p < 0.01$) but *Gender* was not ($p > 0.05$). The second-best bibliometric predictor by AIC, number of outputs weighted by number of authors, was also a measure of research quantity. It had correlation $r^2 = 0.280$. By comparison the other bibliometrics, all measures more related to research quality, had $r^2 < 0.09$. We also tested a two bibliometric model by combining the best predictor, $N_{outputs}$, with each of the other bibliometrics separately. In all cases the change in AIC due to the addition of the second bibliometric was very small ($\Delta AIC < 2$) and the secondary bibliometric was not significant ($p > 0.05$) (see **Supplementary file 4**).

The bibliometric analysis shows several things. First, **Figure 2A** shows the best available predictor of research score was number of research outputs; and bibliometrics related to quality (e.g. expected field weighted citation index) had almost no effect. Alone, this result is surprising as documentation on PBRF clearly states that the evaluations are based on quality not quantity of research (**Commission TE, 2017**). Similarly, *Gender* is not a predictor of research quality, a man and a woman of the same age with the same number of research outputs have the same expected score. **Figure 2A** shows the raw scores of all individuals in the single university sample and the predicted relationship between number of outputs and score for a 50-year-old of either gender. On average, men have more outputs than women (**Figure 2A**, vertical error bars) and get higher scores (horizontal error bars).

Having seen that score is strongly correlated to the number of research outputs (**Figure 2A**), we then test the relationship between the number of outputs and the proportion of men in the discipline using the same sample of 384 individuals at UC (**Figure 2B**). We start with the maximal model to allow the relationship to change for each gender

$$log\left(N_{outputs}\right) \sim p_{men} * Gender.$$

The gender balance of the discipline is highly significant (coefficient $p_{men}$: $p < 0.001$) but again *Gender* is not significant either alone or with an interaction ($p > 0.05$). So the most parsimonious model is

$$log\left(N_{outputs}\right) \sim p_{men}$$

meaning a man and woman in the same discipline have, on average, the same number of outputs and researchers in male-dominated disciplines have, on average, more publications. Taken together, it is tempting to use these correlations as an explanation for the lower scores for researchers in female-dominated disciplines. In other words, both men and women in female-dominated disciplines publish less, and fewer publications lead to a lower score. This explanation fails the empirical test because, as seen in *Figure 1*, on average a man receives a higher score than a woman in the same discipline. This is *despite*, on average, having the same number of outputs (*Figure 2B*).

Finally, we test the relationship of our previous analysis relating *Score* to *Gender* and gender balance, $p_{men}$, but now we add the best bibliometric predictor $N_{outputs}$

$$Score \sim Age + Gender * p_{men} + log\left(N_{outputs}\right).$$

After using this model to account for the difference in publication norms across disciplines with different gender balance, the proportion of men in the discipline is still a significant predictor of score coefficient and interaction $p < 0.05$, see *Supplementary file 4*, but now the relationship is more intricate.

In a female-dominated ($p_{men} = 25\%$) discipline, like Nursing or Education, the expected number of outputs is low for both genders (*Figure 2B*) and scores are correspondingly low. But given that a man and a woman have the same expected number of outputs for their discipline (as predicted by *Figure 2B*), the man's expected score of 412/700 is higher than the woman's of 361/700 (*Figure 2C*, left bars). When we consider a male-dominated ($p_{men} = 75\%$) discipline, like Physics or Philosophy, given gender-equal (but now likely to be high) publication rates, overall scores are much higher, a man's score is predicted to be 467/700 and a woman's is higher still at 505/700 (*Figure 2C*, right bars). In short, even after accounting for variations in publication rates, researchers in female-dominated disciplines still have lower expected research scores.

In this detailed look at PBRF scores compared to bibliometric measures, the gender minority has an advantage over the majority at both ends of the gender balance spectrum. If those in the gender minority of a discipline have the same number of publications as their majority colleagues (as our findings suggest they usually do), the gender minority will, on average, receive a higher research evaluation score.

Here, we have presented a more detailed analysis which accounts for publication differences. Whilst differences between the gender minority and majority in a discipline continue to exist, the key point remains: when women are in the majority, the scores are lower for everyone (*Figure 2C*).

Far more men work in male-dominated disciplines (PBRF 2018: 74% of men in disciplines with $p_{men} > 50\%$) than women (PBRF 2018: 47% of women in disciplines with $p_{men} > 50\%$). Hence, men will benefit overall from a pattern of evaluating research in male-dominated disciplines as higher quality than research in female-dominated disciplines.

## Results

### Research funding success rates

We examine funding success rates using three independent datasets. *Table 1* describes the datasets and shows summary statistics including the number of individuals each dataset covers and the resulting sample size. The raw data in *Table 1* suggest women have lower success rates than men across all the datasets.

We use three independent datasets from: the ARC, the CIHR, and the EIGE. Each dataset gives aggregated data on the number of applicants and the number of successes, by gender and discipline, for research funding. For the ARC and CIHR this is over a number of years, for EIGE this is in a single year. For EIGE the gender balance of each discipline was available separately for over 80% of the dataset and covered all researchers in the discipline nationally, in the remaining cases it was approximated with the applicant gender balance. For ARC gender balance was available separately (see Supplementary Materials for details) and again covered the whole population though it was only for a single year (2018). Data is also available for 2015 but as no discipline changed by more than 2 percentage points between 2015 and 2018 it was not used here. For CIHR we used an approximation based on the number of applicants.

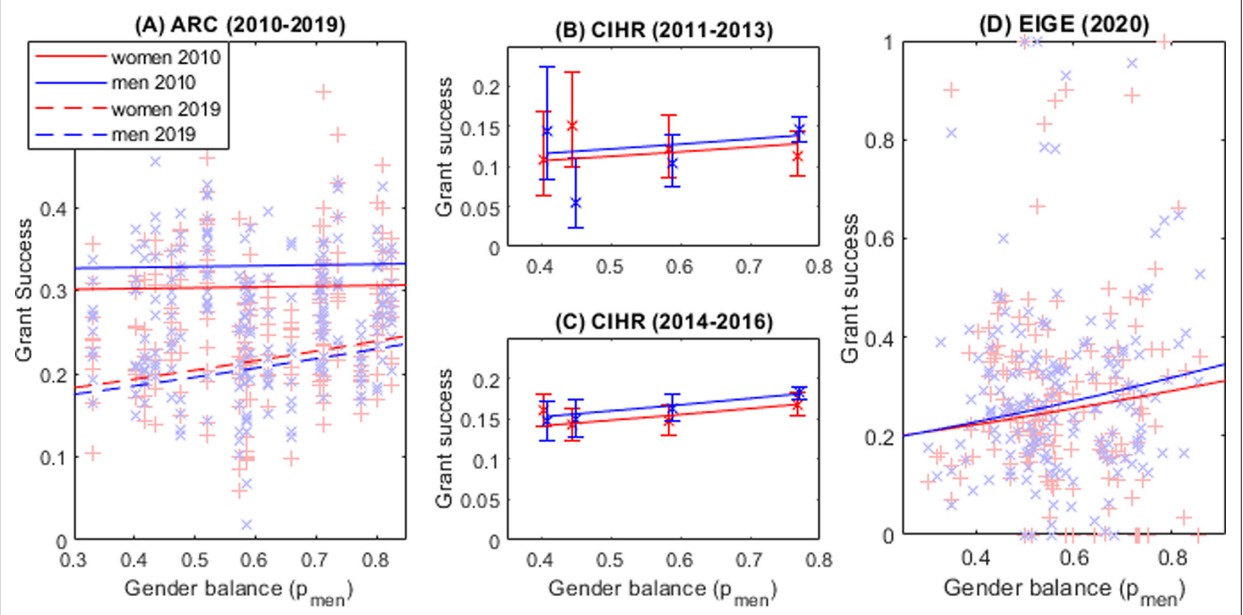

**Figure 3.** Researchers in male-dominated disciplines have a higher chance of funding success. (**A**) Australian Research Council (ARC) (2010–2019): Points – funding success rates by gender in 20 disciplines over 10 years against proportion of men in the discipline in 2018. Lines – expected success rates of men and women in 2010 (solid lines) and 2019 (dashed lines). (**B**) Canadian Institute of Health Research (CIHR) (2011–2013), (**C**) CIHR (2014–2016): Points – funding success rates by gender for each discipline against proportion of men in the discipline (estimated from application numbers) error bars are binomial 95% CI. Lines – expected success rate from combined analysis of all grant types in both time periods. (**D**) European Institute of Gender Equality (EIGE) (2020): Points – funding success rates by gender in 8 disciplines and 27 countries against proportion of men in the discipline in that country. Lines – expected success rates of men and women.

We analyse each dataset separately and use a logistic regression model to predict the funding success rate. As our three datasets vary considerably rather than choose a model a priori we start with a maximal model that includes the key variables *Gender*, $p_{men}$, and, when available, *Time* with up to two-way interactions between all variables. Any additional available variables are included as linear terms (see *Table 2*). We compare all candidate models up to the maximal model and choose by minimising AIC. Analysis was done on raw data so each group was weighted by sample size. Detailed data descriptions are given in Supplementary Materials.

## Australian Research Council

The data included applicant numbers and successes across 22 disciplines, corresponding to the Australian Fields of Research Classifications, over the 10 years from 2010 to 2019. See Supplementary Materials for a more detailed data description. Full output from all models tested is in *Supplementary file 3* and data are available in *Source data 1*. The best fit model was

$$logit\left(P\left(success\right)\right) \sim \left(gender + p_{men}\right) * Year.$$

All predictors and interactions are highly significant ($p < 0.001$, see Model ARC2 in *Supplementary file 3*).

In 2010, there was a fairly constant disadvantage for women. Gender balance of the discipline had a relatively minimal effect on success; e.g., in female-dominated ($p_{men} = 25\%$) disciplines like Nursing a man's chance of success was 32.6% and hers 30.1%. Similarly, in a male-dominated discipline ($p_{men} = 75\%$) like Philosophy a man's success rate would have increased marginally to 33.1% and a woman's to 30.5% (solid lines, *Figure 3A*). In 2010, women's disadvantage in grant applications was affected more by their gender than by their discipline.

By 2019, overall success rates were much lower (dashed lines *Figure 3A*). The effect of gender was now reversed, giving marginally higher success rates to women but only when compared to men in the same discipline. In other words, a male Psychology researcher ($p_{men} = 50\%$) has a 19.6% success rate compared to a female researcher in the same field at 20.4%. But now the discipline's gender balance

has a much larger effect. A male Philosopher's ($p_{men}$ = 75%) success rate has increased to 22.4% and a female Philosopher's success rate is even higher at 23.3%. However, in Nursing ($p_{men}$ = 25%), a female applicant has only a 17.8% probability of success; this is slightly higher than the male applicant's 17.0% probability of success. Compared to 2010, the gender disadvantage has been replaced by the larger disadvantage of being in a female-dominated discipline.

## Canadian Institute of Health Research

This dataset was published in *Witteman et al., 2019*, and includes the number of applications and success rates by gender from 23,000 applicants to the CIHR in four sub-disciplines of Medical Science (Clinical, Biomolecular, Public Health, and Health Sciences). However, this is an aggregated data so the number of data points is much smaller (see *Supplementary file 1*). The time period used was before and after a change in funding model in 2013. See Supplementary Materials for a more detailed data description. Full output from all models tested is in *Supplementary file 3* and data is available in *Source data 2*. The best model (Model CIHR2 in *Supplementary file 3*) was

$$logit\left(P\left(Success\right)\right) \sim Time + Gender + p_{men}.$$

The original study, which accounted for age, found the discipline with the highest proportion of men (Biomedical, 77% men) had significantly higher odds of success than the three other disciplines, with 41–59% men (*Witteman et al., 2019* at Tables 2 and 3). We do not have the age data controlled for in the original study, but we do find that there is a significant relationship ($p < 0.001$) between grant success and the proportion of men in the discipline. Unlike the ARC data men are slightly more likely to be successful overall. In both time periods Canadian health sub-disciplines that are more male-dominated have a higher chance of funding success (before the funding change – *Figure 3B*, after the funding change – *Figure 3C*). Our model predicts that the advantage of working in the most male-dominated discipline (Biomedical, 77% male) compared to the most female-dominated (Public Health, 41% men) is almost three times the size of the direct advantage of being male.

## European Institute of Gender Equality

This dataset contains the number of applications and success rates by gender and country of origin from over 135,000 applicants to government research funds predominantly in the EU but also including the UK, Israel, and Turkey in 2019. Applications are divided into broad research areas, e.g., Science, Engineering, Humanities, etc. Data was not available for all eight disciplines in every country. See Supplementary Materials for a more detailed data description. Full output from all models tested is in *Supplementary file 3* and data is available in Source data file 3. The best model (Model EIGE3 in *Supplementary file 3*) was

$$logit\left(P\left(Success\right)\right) \sim Country + Gender * p_{men}.$$

All variables are significant ($p < 0.05$). In male-dominated fields women have a slightly lower success rate in male-dominated disciplines and, as with the other funding datasets, there is an overall increase in success rates as the discipline becomes more male-dominated. Once again, the effect on success rate of the applicant's gender is small compared to the effect of the gender balance of the applicant's discipline (*Figure 3D*). In the most male-dominated disciplines (80% male) a man has a 2 percentage point advantage over female colleagues; by comparison a man has an 11 percentage point advantage over all researchers working in the most female-dominated discipline (20% female). In relative terms this results in researchers in male-dominated disciplines being 50% more likely to be successful in funding applications than researchers in male-dominated disciplines.

## Discussion

We present two similar, yet separate, findings: (1) gender balance in a discipline correlates with research quality scores; and (2) gender balance in a discipline correlates with research funding success rates. These findings do not identify a causal relationship in which gender balance causes low research scores or funding success or vice versa. But we can highlight and explore some possible explanations

and theories that might link gender balance and research score/funding success. Further, even without establishing causality, these paired findings offer another piece of the academic gender puzzle.

## Feminisation

Some have found that as a workforce feminises, salaries fall (*England et al., 2007*). To explore whether as a discipline feminised the evaluation of research quality in that discipline also changed, we would need extensive longitudinal data. We cannot do that with the available data. But it is possible that the finding that research in female-dominated disciplines is evaluated as less good than research in male-dominated disciplines is an outcome of feminisation over time.

## Individual gender bias

One possible explanation of our findings is simply that reviewers (of research portfolios in NZ, and of research proposals in Australia, Canada, and the EU and UK) are biased against women. Indeed the effects of unconscious bias on academics are theorised thoroughly (*Poppenhaeger, 2019*; *Pritlove et al., 2019*). Although empirics are limited (*Skov, 2020*) it has been seen that unconscious biases can affect common academic metrics and practices such as h-index, citation, authorship, and peer review, as well as hiring and progression (*Kieokaew and IDEEA collaboration, 2023*).

The PBRF datasets offer the most granular data and hence the best opportunity to explore gender bias on an individual level. For example, if the PBRF was dominated by individual bias this would manifest itself in Education having lower average scores than Physics but a woman in Education would have the same score as a woman in Physics on average. We see the former, but not the latter. However, we do see that across all disciplines women have lower scores than men and this is more pronounced in female-dominated disciplines which is suggestive of gender bias but far from conclusive due to the many factors at play.

Further, the bibliometric subnational dataset suggests that a man and a woman with the same number of outputs (the best bibliometric predictor of score) have the same expected score (*Figure 2B*). But when we include discipline gender bias (*Figure 2C*) the picture is less clear suggesting that if we compare a man and a woman with the same number of outputs the person in the minority gender is likely to have a higher score.

## Women do lower quality research

A small elephant in the room that might explain our findings is simply that women do lower quality research than men. Our findings do not show this. In 2018 in a male-dominated subject like Physics, a woman's score is expected to be the same as a man's (*Figure 1*). Similarly, an Australian woman has a slightly higher funding rate than a man in the same discipline in the ARC. These point to men and women having very similar research quality after accounting for discipline.

We do see that researchers, both male and female, in female-dominated disciplines have fewer research outputs on average. This could be due to cultural norms in these disciplines being driven by women who overall have fewer outputs; or it could be that women are attracted to disciplines with lower publication norms.

## Pre-allocation of and quotas for research funds

For research funding datasets (ARC, CIHR, EIGE), quotas or pre-allocation of funds towards certain types of research (e.g. public health funding increasing during the Covid pandemic) could explain some of our findings. But these pre-allocations would need to be towards male-dominated disciplines. Such a gendered pattern of pre-allocation suggests it would be more of a different mechanism for our findings, rather than an explanation of them. But we examine the datasets in turn against this explanation.

Most ARC applications are to the Discovery Project fund. Here, applications are scored independently and there are no pre-allocation decisions to value or fund one discipline over another (*Meho, 2021*). Application success rates in 2023 are similar across the five very broad fields of research. This supports ARC's claim that the process does not overtly pre-allocate by discipline. However, when we consider applicant success rates over the more detailed 22 disciplines, female-dominated disciplines have lower success rates.

There is little detailed data about the underlying processes that lie behind the EIGE data. Every country has its own internal rules and priorities for funding allocation. If local policy prioritises funding to Engineering and Physical Science, this could result in higher funding success rates in these areas; although it could also result in these areas having more researchers and the success rate being unaffected. Similarly we have no information on the Canadian funding system.

Conversely, in PBRF, this effect of pre-allocation is negligible or even non-existent. Every individual is scored independently. A surfeit of high scoring individuals in Physics does not stop high scores in Education. Overall, the pre-allocation of funds cannot explain all our results; and our results appear robust to this explanation.

## Job choice

This is frequently offered as an explanation of the gender pay gap. Women choose to work in job fields that, coincidently, are paid less (*Baker and Fortin, 2001*). It is a possibility that discipline choice, as opposed to job choice, is at play here, where women choose to work in disciplines where research is seen as less good. In academia, women tend to choose research areas and academic disciplines that the educational system undervalues (*Key and Sumner, 2019*). Additionally, within a discipline, women tend to choose research topics that are more interdisciplinary and more applied (*Kim et al., 2022*).

Several individual and contextual factors affect job choice and evaluations (*Carli et al., 2019*). Perceptions of excellence can correspond with an underrepresentation of women in certain disciplines, resulting in a higher perceived genius status for that discipline (*Leslie et al., 2015*). Our results find a correlation but not the causation for female-dominated disciplines having lower evaluations of research quality. Hence, we can neither support nor reject an explanation of job choice.

## Bias against female-dominated disciplines

A simple explanation for our findings is that evaluations of research are biased but not against individuals as considered above earlier. Rather this explanation posits a bias against research disciplines dominated by women. Again as our findings are correlation not causation, we cannot accept or reject this explanation with certainty. Instead we can point to other research that suggests that we perceive male-dominated disciplines to require 'brilliance', while female-dominated disciplines require 'hard work' (*Leslie et al., 2015*). If these discipline-specific perceptions spill over into evaluations of research and funding applications, it could produce a gender-based discipline bias very similar to our findings.

Another simple explanation is that research in female-dominated disciplines is of lower quality than research in male-dominated disciplines. Research varies widely across disciplines from the quantitative 'hard sciences' to the qualitative 'soft sciences' but to categorise the efforts of whole disciplines as lower quality rather than different is an extreme step. This explanation also raises the question of why disciplines perceived as lower quality are more likely to be female-dominated.

Speculation aside, our findings do not show that research done in disciplines dominated by women is of lower quality. They do suggest that we consistently evaluate research from female-dominated disciplines as lower quality than research from male-dominated disciplines. This has a detrimental effect on everyone in those disciplines, but will adversely affect more women than men.

These findings will have implications in labour environments where research funding or research score affect salary or rank of individuals (*Brower and James, 2020*). Future research could include modelling of various 'levers of change' (*James and Brower, 2022*) that might serve to diminish, correct, or compensate for the gendered patterns observed.

Our datasets do have limitations. They cover a range of countries but are not representative of all of academia worldwide. And neither research evaluation nor funding success is a perfect measure of academic quality.

## Conclusion

We present two separate findings: gender balance of the discipline is strongly related to research quality evaluations and to funding success. We cannot claim to establish causality nor combine the two findings. Instead we seek possible explanations for our findings in others' work.

Our findings suggest that what is perceived as women's research is valued less, whether it is a man or a woman doing the research and whether or not overt bias is to blame. Regardless of causality, our

findings suggest that patterns of devaluing women's work affect all who do it, regardless of gender. And our findings point to areas that are ripe for further exploration.

## Acknowledgements

We thank Pen Holland, Elena Moltchanova, and Ximena Nelson for providing comments on a draft of this work.

## Additional information

### Funding

| Funder | Grant reference number | Author |
|---|---|---|
| Bio-Protection Research Centre | | Franca Buelow |

The funders had no role in study design, data collection and interpretation, or the decision to submit the work for publication.

### Author contributions

Alex James, Conceptualization, Data curation, Formal analysis, Investigation, Visualization, Methodology, Writing - original draft, Project administration, Writing - review and editing; Franca Buelow, Methodology, Writing - original draft, Writing - review and editing; Liam Gibson, Investigation, Methodology; Ann Brower, Conceptualization, Investigation, Visualization, Methodology, Writing - original draft, Writing - review and editing

### Author ORCIDs

Alex James (ID) https://orcid.org/0000-0002-1543-7139

Reviewer #2 (Public Review): https://doi.org/10.7554/eLife.97613.3.sa1
Reviewer #3 (Public Review): https://doi.org/10.7554/eLife.97613.3.sa2
Author response https://doi.org/10.7554/eLife.97613.3.sa3

## Additional files

### Supplementary files

• Supplementary file 1. Table S1: Comparative descriptions of each of the four datasets.

• Supplementary file 2. Table S2: Summary statistics of the Performance-Based Research Fund (PBRF) data by discipline. Includes the number of women and men, mean age, and mean score for each gender and proportion of men in each discipline.

• Supplementary file 3. Detailed model output for all Performance-Based Research Fund (PBRF), Australian Research Council (ARC), Canadian Institute of Health Research (CIHR), and European Institute of Gender Equality (EIGE) data candidate models. Each sheet shows the coefficients and p-values for the full range of candidate models trialled for each dataset to predict either Score (PBRF) or Funding success (ARC, CIHR, EIGE). Coefficient names (first column) are given in Wilkinson's notation. ΔAIC is the change in AIC between the best fit model (minimum AIC, ΔAIC = 0) and the candidate model. r-Squared is also given for the linear model (PRF only). Individual country coefficients and p-values for the EIGE data are not shown for clarity.

• Supplementary file 4. Detailed model output for the bibliometric models. Sheet 1. Single bibliometric models $Score \sim Age + Gender + log\left(Bib\right)$. The coefficients and p-values for each of the five bibliometrics. ΔAIC is the change in AIC between the best fit model, $N_{outputs}$ with minimum AIC and ΔAIC = 0, and the candidate model. r-Squared is also given. Sheet 2. Two bibliometric model $Score \sim Age + Gender + log\left(N_{outputs}\right) + log\left(Bib\right)$. The coefficients and p-values for each of the other four bibliometrics. ΔAIC is the change in AIC between the $N_{outputs}$ only model (minimum AIC, ΔAIC = 0) and the candidate model. r-Squared is also given. Sheet 3. Coefficients and p-values for $Score \sim Age + Gender * p_{men} + log\left(N_{outputs}\right)$.

- MDAR checklist
- Source data 1. DataARC.xlsx. Field, Gender, Year, Number of applicants, Number of successes, proportion men (from external data).
- Source data 2. DataCIHR.xlsx. Field, Gender, Type of grant, Time period (before or after funding change), Number of applicants, number of successes, proportion male (from application numbers).
- Source data 3. DataEIGE.xlsx. Country, Field, Gender, Number of applicants, Number of successes, proportion male (from applicant data), proportion male from external data.

## Data availability

All publicly available data analysed during this study are included in the manuscript and supporting files. The NZ PBRF data is not publicly available but can be requested from the NZ TEC via the Official Information Act.

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

# Appendix 1

## Supplementary materials

### Detailed data descriptions

#### ANZ: PBRF research evaluation scores

New Zealand's PBRF's assessment scores every research-active university academic's research performance from 200 to 700 points using a personal portfolio. Assessments are holistic and purport to be based more on quality and impact than sheer number of publications (1). A portfolio contains full texts and impact summaries of an academic's nominated four best publications (books, articles, exhibitions, etc.), a list of additional publications and postgraduate supervisions, and evidence of the academic's contribution to the research environment and peer esteem (journal editorial posts, speaking invitations, awards, etc.) during the 6-year assessment period. PBRF takes part-time employment and special circumstances (illness, heavy administrative load, etc.) into account.

Researchers are assessed by one of fourteen panels consisting of national and international experts grouped by research field; e.g., political science and geography disciplines are both grouped under the Social Sciences panel. Panels review their preliminary scores for evidence of patterns of bias before moderating and finalising the scores.

Assessments give each individual academic three component scores: research outputs, peer esteem, and contribution to the research environment. These are combined in the ratio 70:15:15 to give an overall score from 0 to 700. Scores are then clustered into grades (600–700=A; 400–599=B; 200–399=C; 0–200=R [research inactive]) which are reported back to individuals. Research-inactive individuals were not included in this study.

**Appendix 1—table 1.** A summary of the three Performance-Based Research Fund (PBRF) datasets. Mean (and standard deviation) of score and age at each time point. ***p<0.001, *0.01<p<0.05, two-sided t-test men/women.

| Time period | | 2006 | 2007–12 | 2013–18 |
|---|---|---|---|---|
| Disciplines | | 42 | 42 | 43 |
| *N* | Women | 1708 | 2658 | 3297 |
| | Men | 2522 | 4005 | 4181 |
| Mean score (std) | Women | 272 (161)*** | 368 (139)*** | 379 (142)*** |
| | Men | 351 (163) | 424 (143) | 433 (146) |
| Mean age (std) | Women | 44.5 (9.6)* | 47.8 (10.4)*** | 47.7 (11.1)*** |
| | Men | 45.3 (10.1) | 50.0 (10.9) | 49.5 (11.4) |

### Discipline groupings

Individuals self-allocated to 1 of 42 disciplines covering all fields of research at very high level of detail (see *Supplementary file 1*). At the third assessment in 2018 an additional discipline was added to give a total of 43.

### Gender balance

As the dataset is the entire population of active researchers the proportion of men in each discipline is taken directly from the data.

### Time

The dataset covers three time spans (2000–06, 2007–12, 2013–18). We chose to analyse the three time points separately, i.e., treat them as independent datasets. This was due to the relatively long time between assessments and the large number of data points at each time.

### Other available data

The PBRF dataset also contains detailed data on the research institution of each individual, i.e., one of the eight ANZ universities; researcher date of birth; researcher job title and academic rank, i.e., Professor, Associate Professor, etc.; researcher ethnicity. In our analysis we accounted for age and institute as continuous and categorical variables respectively. We chose not to include ethnicity as many of the ethnicity groupings were only represented in a small number of disciplines. Job title or academic rank is in itself a measure of research quality albeit a very implicit one with many

confounding factors. It has also been shown repeatedly to be subject to many gender biases through promotion and hiring biases. For this reason we did not include it but rather relied on age as a proxy for seniority and experience.

## Sample size

The final sample sizes for the three sub-datasets of PBRF are 2006: 4230 data points; 2012: 6663 data points; 2018: 7487 data points. See *Table 1* for the gender breakdown and average research quality of each group.

## Independence

The three sub-datasets are not strictly independent as many individuals will have been employed at all three assessments. However, at a single assessment all individuals were assessed independently. There were no pre-allocations of scores to different disciplines, i.e., a preponderance of very high scores in one discipline did not require another discipline to be down-graded. Results from the 14 broad subject area panels were cross-referenced with each other to standardise across disciplines.

## Availability

The data used in this study are owned by a third-party organisation (Tertiary Education Commission [TEC], New Zealand). The authors were granted access privileges to the data, under strict nondisclosure agreements, by the TEC for this research project only, under ANZ's Official Information Act 1992 which facilitates New Zealanders' access to government records, through a formal information request. All ANZ citizens and residents may make such requests, under the following guidelines: https://www.dia.govt.nz/Official-Information-Act-requests. This dataset pertains to thousands of people's employment; hence it is strictly private and highly sensitive. Due to ethical and privacy restrictions, a de-identified dataset cannot be made publicly available. Interested researchers are invited to contact the corresponding authors to discuss access to data. Methods (*Supplementary file 1*) contains summary data by discipline for each time point.

## UC bibliometric data

We matched the anonymised PBRF data with the UC internal research database using date of birth, gender, ethnicity, discipline, and academic department. Conclusive matches were found for 384 individuals. Researchers with no recorded research outputs were not included. Publications data includes journal publications, books, book chapters, and technical reports published during the PBRF assessment period 2012–18. The publication details of each individual output were combined with information from Scopus about publication outlet, citations (counted in February 2022), and number of co-authors. Where possible, for each individual we calculated the five bibliometrics in *Supplementary file 4*.

## Discipline groupings

Defined by each individual's 2018 PBRF discipline.

## Gender balance

The gender balance of the entire ANZ researcher workforce from the full 2018 PBRF dataset.

## Time

Research score was the 2018 research quality score. Publications were all those recorded between 2013 and 2018, inclusive, the approximate dates of the PBRF assessment.

## Bibliometric data

The internal database gave a record for every research output by each individual. A research output record included: publication source, e.g., journal name, book publisher, or report name; Source normalised impact per paper (SNIP – a measure of the journal impact relative to the field); co-author information; citations received by February 2022; field weighted citation impact (FWCI – a measure of the number of citations received with reference to the average number for that field). Research outputs not in journals did not have a SNIP value so were excluded from some calculations. This resulted in five bibliometric variables for each individual. The first two are a measure of the quantity of outputs by a researcher during the assessment period. The final three are more heavily related to the quality of an individual's work.

$N_{outputs}$ – total number of outputs for that researcher.

$N_{weighted}$ – total number of outputs weighted by number of co-authors, i.e., a single author publication counts 1, a two author publication counts ½, etc.

$E\left(FWCI\right)$ – mean FWCI across the individual's publications.

$E\left(SNIP\right)$ – mean SNIP across the individual's publications.

·$E\left(Cites\right)$ – mean number of citations per publication by that individual.

## Sample size

384 individuals were able to be matched with their PBRF data. In total there were 7689 publication records.

## Independence

Although the dataset did not include the entire research-active population of UC researchers there was no bias as the smaller sample was due to matching problems centred around unreliable dates of birth in the two datasets.

## Availability

Although research publications of staff at the UC are publicly available the data linked to the PBRF score of each individual remains confidential as described above.

## ARC funding success data

Funding success data by gender and field of research for applications to the ARC are publicly available at https://www.arc.gov.au/sites/default/files/arc_ncgp_gender_trend_data_for_web_nov2019.xlsx (see Table 4; Field of Research). We used data from 2010 to 2019 on all grant applications by two-digit Field of Research code as assigned by the applicants. Data included all funding schemes in the National Competitive Grants Programme finalised and announced by the Minister for each year. The number of applicants includes all named researchers on the application, including partner investigators.

## Discipline groupings

ARC uses 22 Fields of Research that give a relative fine categorisation of research discipline. Researchers self-allocated to a discipline grouping.

## Gender balance

The dataset only contains the gender breakdown of individuals that applied for funding. This is likely to be a biased sample of the researcher population. Instead, we use data from the 2018 Survey of the Australian research population. This gives the number of researchers in each of the 22 disciplines across all Australian universities broken down by gender in 2018.

## Time

The dataset covers the 10-year period from 2010 to 2019 (inclusive). As the dataset is smaller than the PBRF dataset we choose to treat it as a single dataset with a continuous time variable.

## Other available data

The 2018 researcher survey also gives a rank breakdown of researchers by gender and discipline. However, as in the case of PBRF we see this variable as being too confounded with grant success to use as an additional predictor variable.

## Sample size

The data is not at the level of an individual. It is aggregated data giving the number of applicants and the number of successes in each discipline each year. The data is split by gender. In total there are 440 (=22 disciplines × 10 years × 2 genders) independent data points that include over 176,000 applicants. Note that one individual can apply multiple times in multiple disciplines with varying success, generating multiple 'applicants'.

## Independence

In the dominant grant categories (Discovery grants) the ARC does not pre-allocate funds to different disciplines during its decision process. Applications are scored and ranked individually to decide funding success.

## Availability

These are publicly available at https://www.arc.gov.au/sites/default/files/arc_ncgp_gender_trend_data_for_web_nov2019.xlsx and https://dataportal.arc.gov.au/era/nationalreport/2018/pages/section1/gender. The subset of the data used in this project are available in *Source data 1*.

## CIHR funding success data

The data are from a previous published analysis by *Witteman et al., 2019* (their Table 2) on the success rates of over 23,000 funding applicants to the CIHR.

### Discipline groupings

This dataset contains only four disciplines which are all sub-disciplines of medical science (Public Health, Bio-molecular, Clinical and Health Sciences). Researchers self-allocated to a discipline grouping.

### Gender balance

There was no available data on the gender balance of the researcher population. Instead we used the gender balance of the applicants. As stated previously, this is likely to be biased but unfortunately was the best available proxy.

### Time

The data is available at two time points: before and after a change in funding processes (i.e. before 2013 there was a single grant category – traditional, this split into foundation and project grants after 2013). The original study found a difference in the success rates of women before and after the change. To account for this we continue to split the data into the two time points and treat this as a categorical variable.

### Other available data

The original study accounted for researcher age but this aspect of the data is not publicly available. The original study split the post-2013 data into two types of grant application (project and foundation). These were aggregated here as this was not relevant to our study.

### Sample size

The data is not at the level of the individual. It is aggregated data giving the number of applicants and the number of successes in each discipline at each time point. The data is split by gender. There are 16 (=4 disciplines × 2 time points × 2 genders) independent data points that include over 23,000 applicants. As with the ARC data one individual can apply multiple times in multiple disciplines with varying success, generating multiple 'applicants'.

### Independence

No information was available on the independence of the funding allocation. Pre-allocation decisions could have been made to give proportionally higher success rates to some disciplines at a cost to others.

### Availability

Our study did not use the original dataset described by *Witteman et al., 2019*. The data used are from analysis of this data and are published in Table 2 and Table 3 of *Witteman et al., 2019*, and summarised in *Source data 2*.

## EIGE funding success data

The data are the funding success rates of government-funded grant applications from the EU and UK from EIGE (*Hofstra et al., 2020*). Each country is recorded separately.

### Discipline groupings

The discipline groupings are much coarser than the other datasets. It uses eight disciplines that each cover a broad area, e.g., Science, Humanities, Engineering, etc. Methods for allocation to particular disciplines are not available.

### Gender balance

A separate dataset (*Ding et al., 2006*) gives the gender breakdown of the same disciplines by country. In a small number of country/discipline combinations there is no population estimate of the gender balance. In these cases the gender balance was estimated using the application gender balance.

### Time

The dataset is for a single time point 2019.

### Other available data

There were no other available relevant data.

### Sample size

The data is not at the level of an individual. It is aggregated data giving the number of applicants and the number of successes in each discipline/country combination. The data is split by gender. In total there are up to 448 (=8 disciplines × 28 countries × 2 genders) independent data points that include over 135,000 applicants. Note that one individual can apply multiple times in multiple disciplines with varying success, generating multiple 'applicants'. After missing data is removed there are 333 data points. Of the 333 available data points 47 (14%) have no available gender balance and gender balance of the applications is used as a proxy.

## Independence

No information was available on the independence of the funding allocation. Pre-allocation decisions could have been made to give proportionally higher funds to some disciplines at a cost to others.

## Availability

These are publicly available at Indicator: Research funding success rate differences (percentage points) between women and men, by field of R&D and in total | Gender Statistics Database | European Institute for Gender Equality (europa.eu) (https://eige.europa.eu/gender-statistics/dgs/indicator/ta_resdig_sctech_funding_sf_fund_appl_ben/datatable) and p_men taken from https://eige.europa.eu/gender-statistics/dgs/indicator/ta_resdig_sctech_rdperes_perf_rd_%2op_perssci_prop. Data is collected by EIGE based on the 2021 She Figures publication, based on the women in science (WiS) questionnaire module T1. The subset of the data used in this project are available in *Source data 3*.

All analysis was done with Matlab (2022b), primarily using the glmfit function.

