## [Editor Report · eLife assessment]

This study provides **convincing** evidence that the quality of research in female-dominated fields of research is systematically undervalued by the research community. The authors' findings are based on analyses of data from a research assessment exercise in New Zealand and data on funding success rates in Australia, Canada, the European Union and the United Kingdom. This work is an **important** contribution to the discourse on gender biases in academia, underlining the pervasive influence of gender on whole fields of research, as well as on individual researchers.

---

## [Referee Report · Reviewer #2 (Public Review)]

Summary:

The authors used four datasets spanning 30 countries to examine funding success and research quality score for various disciplines. They examined whether funding or research quality score were influenced by majority gender of the discipline and whether these affected men, women, or both within each discipline. They found that disciplines dominated by women have lower funding success and research quality score than disciplines dominated by men. These findings are surprising because even the men in women-dominated fields experienced lower funding success and research quality score.

Strengths:

- The authors utilized a comprehensive dataset covering 30 countries to explore the influence of the majority gender in academic disciplines on funding success and research quality scores.

- Findings suggest a systemic issue where disciplines with a higher proportion of women have lower evaluations and funding success for all researchers, regardless of gender.

- The manuscript is notable for its large sample size and the diverse international scope, enhancing the generalizability of the results.

- The work accounts for various factors including age, number of research outputs, and bibliometric measures, strengthening the validity of the findings.

- The manuscript raises important questions about unconscious bias in research evaluation and funding decisions, as evidenced by lower scores in women-dominated fields even for researchers that are men.

- The study provides a nuanced view of gender bias, showing that it is not limited to individuals but extends to entire disciplines, impacting the perception and funding and quality or worth of research.

- This work underscores the need to explore motivations behind gender distribution across fields, hinting at deep-rooted societal and institutional barriers.

- The authors have opened a discussion on potential solutions to counter bias, like adjusting funding paylines or anonymizing applications, or other practical solutions.

- While pointing out limitations such as the absence of data from major research-producing countries, the manuscript paves the way for future studies to examine whether its findings are universally applicable.

- The study carefully uses the existing data (including PBRF funding panel gender diversity) to examine gender bias. These types of datasets are often not readily accessible for analysis. Here, the authors have used the available data to the fullest extent possible.

The authors have addressed the concerns I had about the original version.

---

## [Referee Report · Reviewer #3 (Public Review)]

This study seeks to investigate one aspect of disparity in academia: how gender balance in a discipline is valued in terms of evaluated research quality score and funding success. This is important in understanding disparities within academia.

This study uses publicly available data to investigate covariation between gender balance in an academic discipline and:

individual research quality scores of New Zealand academics as evaluated by one of 14 broader subject panels.

[ii] funding success in Australia, Canada, Europe, UK.

The authors have addressed the concerns I had about the original version

---

## [Author Response]

The following is the authors’ response to the original reviews.

**Public Reviews:**

**Reviewer #2 (Public Review):**
Summary:The authors used four datasets spanning 30 countries to examine funding success and research quality score for various disciplines. They examined whether funding or research quality score were influenced by majority gender of the discipline and whether these affected men, women, or both within each discipline. They found that disciplines dominated by women have lower funding success and research quality score than disciplines dominated by men. These findings, are surprising because even the men in women-dominated fields experienced lower funding success and research quality score.Strengths:- The authors utilized a comprehensive dataset covering 30 countries to explore the influence of the majority gender in academic disciplines on funding success and research quality scores.- Findings suggest a systemic issue where disciplines with a higher proportion of women have lower evaluations and funding success for all researchers, regardless of gender.- The manuscript is notable for its large sample size and the diverse international scope, enhancing the generalizability of the results.- The work accounts for various factors including age, number of research outputs, and bibliometric measures, strengthening the validity of the findings.- The manuscript raises important questions about unconscious bias in research evaluation and funding decisions, as evidenced by lower scores in women-dominated fields even for researchers that are men.- The study provides a nuanced view of gender bias, showing that it is not limited to individuals but extends to entire disciplines, impacting the perception and funding and quality or worth of research.- This work underscores the need to explore motivations behind gender distribution across fields, hinting at deep-rooted societal and institutional barriers.- The authors have opened a discussion on potential solutions to counter bias, like adjusting funding paylines or anonymizing applications, or other practical solutions.- While pointing out limitations such as the absence of data from major research-producing countries, the manuscript paves the way for future studies to examine whether its findings are universally applicable.Weaknesses:- The study does not provide data on the gender of grant reviewers or stakeholders, which could be critical for understanding potential unconscious bias in funding decisions. These data are likely not available; however, this could be discussed. Are grant reviewers in fields dominated by women more likely to be women?- There could be more exploration into whether the research quality score is influenced by inherent biases towards disciplines themselves, rather than only being gender bias.- The manuscript should discuss how non-binary gender identities were addressed in the research. There is an opportunity to understand the impact on this group.- A significant limitation is absence of data from other major research-producing countries like China and the United States, raising questions about the generalizability of the findings. How comparable are the findings observed to these other countries?- The motivations and barriers that drive gender distribution in various fields could be expanded on. Are fields striving to reach gender parity through hiring or other mechanisms?- The authors could consider if the size of funding awards correlates with research scores, potentially overlooking a significant factor in the evaluation of research quality. Presumably there is less data on smaller 'pilot' funds and startup funds for disciplines where these are more common. Would funding success follow the same trend for these types of funds?- The language used in the manuscript at times may perpetuate bias, particularly when discussing "lower quality disciplines," which could influence the reader's perception of certain fields.- The manuscript does not clarify how many gender identities were represented in the datasets or how gender identity was determined, potentially conflating gender identity with biological sex.
**Reviewer #3 (Public Review):**
This study seeks to investigate one aspect of disparity in academia: how gender balance in a discipline is valued in terms of evaluated research quality score and funding success. This is important in understanding disparities within academia.This study uses publicly available data to investigate covariation between gender balance in an academic discipline and:i) Individual research quality scores of New Zealand academics as evaluated by one of 14 broader subject panels.ii) Funding success in Australia, Canada, Europe, UK.The study would benefit from further discussion of it limitations, and from the clarification of some technical points (as described in the recommendations for the authors).
**Recommendations For The Authors:**

**Reviewer #2 (Recommendations For The Authors):**
This is a very nice study as-is. In the following comments, I have mainly put my thoughts as I was reading the manuscript. If there are practical ways to answer my questions, I think they could improve the manuscript but the data required for this may not be available.Are there any data on the gender of grant reviewers or stakeholders who make funding decisions?The research quality score metrics seem to be more related to unconscious bias. The funding metrics may also, but there are potentially simple fixes (higher paylines for women or remove gender identities from applications).

We have included some details about PBRF funding panel gender diversity. These panels are usually more gender balanced than the field they represent, but in the extreme cases (Engineering, Education, Mathematics) they are skewed as would be expected. Panels for other award decision makers was not available.

I wonder if the research score metric isn't necessarily reflecting on the gender bias in the discipline but rather on the discipline itself? Terms like "hard science" and "soft science" are frequently used and may perpetuate these biases. This is somewhat supported by the data - on line 402-403 the authors state that women in male-dominated fields like Physics have the same expected score as a man. Could it be that Physics has a higher score than Education even if Physics was woman-dominated and Education was man-dominated? Are there any instances in the data where traditionally male- or female-dominated disciplines are outliers and happen to be the opposite? If so, in those cases, do the findings hold up?

Overall we would love to answer this question! But our data is not enough. We mention these points in the Discussion (Lines 472-466). We have extended this a little to cover the questions raised here.

How are those with non-binary gender identities handled in this article? If there is any data on the subject, I would be curious to know how this effects research score and funding success.

These data were either unavailable or the sample size was too small to be considered anonymously (Mentioned on Lines 74-76).

A limitation of the present article is a lack of data on major research-producing countries like China and the United States. Is there any data relevant to these or other countries? Is there reason to believe the findings outlined in this manuscript would apply or not apply to those countries also?

We would be very excited to see if the findings held up in other countries, particularly any that were less European based. Unfortunately we could not find any data to include. Maybe one day!

What are the motivations or other factors driving men to certain fields and women to certain fields over others? What are the active barriers preventing all fields from 50% gender parity?

Field choice is a highly studied area and the explanations are myriad we have included a few references in the discussion section on job choice. I usually recommend my students read the blog post at https://www.scientificamerican.com/blog/hot-planet/the-people-who-could-have-done-science-didnt/. It is very thoughtful but unfortunately not appropriate to reference here.

The authors find very interesting data on funding rates. Have you considered funding rates and the size of funding awards as a factor in research score? Some disciplines like biomedical science receive larger grants than others like education.

A very interesting thought for our next piece of work. We would definitely like to explore our hypothesis further.

There are instances where the authors writing may perpetuate bias. If possible these should be avoided. One example is on line 458-459 where the authors state "...why these lower quality disciplines are more likely..." This could be re-written to emphasize that some disciplines are "perceived" as lower quality. Certainly those in these discipline would not characterize their chosen discipline as "low quality".

Well-spotted! Now corrected as you suggest.

Similar to the preceding comment, the authors should use care with the term "gender". In the datasets used, how many gender identities were captured? How many gender identity options were given in the surveys or data intake forms? Could individuals in these datasets have been misgendered? Do the data truly represent gender identity or biological sex?

We know that in the PBRF dataset gender was a binary choice and transgender individuals were able to choose which group they identified with. There was no non-binary option (in defence the latest dataset there is from 2018 and NZ has only recently started updating official forms to be more inclusive) and individuals with gender not-stated (a very small number) were excluded. ARC did mention that a small number of individuals were either non-binary or gender not stated, again these are not included here for reasons of anonymity. This is now mentioned on Lines 74-76. The effects on this group are important and understudied likely because, as here, the numbers are too small to be included meaningfully.

**Reviewer #3 (Recommendations For The Authors):**
Major revisions:Could you add line numbers to the Supplementary Materials for the next submission?

Yes! Sorry for the omission.

(1) In the main text L146 and Figure 1, it is not clear why the expected model output line is for a 50 year old male from University of Canterbury only, but the data points are from disciplines in all eight universities in New Zealand. I think it would be more clear and informative to report the trend lines that represent the data points. At the moment it is hard to visualise how the results apply to other age groups or universities.

As age and institution are linear variables with no interactions they are only a constant adjustment above or below this line and the adjustment is small in comparison to the linear trend. Unfortunately, if they were included graphically they do not aid understanding. We agree that indluded raw data with an adjusted trend line can be confusing buy after a lor of between-author discussion this was the most informative compromise we could find (many people like raw data so we included it).

(2) Does your logistic regression model consider sample size weighting in pmen? Weighting according to sample sizes needs to be considered in your model. At the moment it is unclear and suggests a proportion between 0 and 1 only is used, with no weighting according to sample size. If using R, you can use glm(cbind nFem, nMalFem).

Yes. All data points were weighted by group size exactly as you suggest. We have updated the text on Lines 317 to make this clear.

(3) For PBRF, I think it is useful to outline the 14 assessment panels and the disciplines they consider. Did you include the assessment panel as an explanatory variable in your model too to investigate whether quality is assessed in the same manner between panels? If not, then suggest reasons for not doing so.

We have now included more detail in main text on the gender split of the panels. They were not included as an explanatory variable. In theory there was some cross-referencing of panel scores to ensure consistency as part of the PBRF quality assurance guidelines.

(4) There are several limitations which should be discussed more openly:Patterns only represent the countries studied, not necessarily academia worldwide.

Mentioned on Line 485-487.

Gender is described as a binary variable.

Discussed on Line 74-76.

The measure of research evaluation as a reflection of academic merit.

This is acknowledged in the data limitations paragraph in the discussion, at the end of the discussion

Minor revisions:(1) L186. Why do you analyse bibliometric differences between individuals from University of Canterbury only? It would be helpful to outline your reasons.

Although bibliometric data is publicly available it is difficult to collect for a large number of individuals. You also need some private data to match bibliometrics with PBRF data which is anonymous. We were only able to do this for our own institution with considerable internal support.

(2) How many data records did you have to exclude in L191 because they could not be linked? This is helpful to know how efficient the process was, should anyone else like to conduct similar studies.

We matched over 80% of available records (384 individuals). We have mentioned this on Line 194.

(3) Check grammar in the sentence beginning in L202.

Thank-you. Corrected.

(4) Please provide a sample size gender breakdown for "University of Canterbury (UC) bibliometric data", as you do for the preceding section. A table format is helpful.

Included on Line 194.

(5) L377 I think this sentence needs revision.

Thank you, we have reworked that paragraph.

(6) L389-392 Is it possible evaluation panels can score women worse than men and that because more women are present in female-biassed disciplines, the research score in these are worse? Women scoring worse between fields, may be a result of some scaling to the mean score.

No. This is not possible because women in male-dominated fields score higher.

(7) L393 Could you discuss explanations for why men outperform women in research evaluation scores more when disciplines are female dominated?

Unfortunately, we don’t have an explanation for this and can’t get one from our data. We hope it will be an interesting for future work.

(8) Could the figures be improved by having the crosses, x and + scaled, for example, in thickness corresponding to sample size? Alternatively, some description of the sample size variation? Sorting the rows by order of pmen in Table E1 would also be helpful for the reader.

As with the previous figure we have tried many ways of presenting it (including tis one). Unfortunately nothing helped.

We have provided Table E1 as a spreadsheet to allow readers to do this themselves.

(9) Please state in your methods section the software used to aid repeatability.

This is now in Supplementary Materials (Matlab 2022b).

(10) It is great to report your model findings into real terms for PBRF and ARC. Please can you extend this to CIHR and EIGE. i.e. describing how a gender skew increase of x associates with a y increase in funding success chance.

We have added similar explanations for both these datasets comparing the advantage of being male with the advantage of working in a male dominated discipline.

(11) I would apply care to using pronouns "his" and "her" in L322-L324 and avoid if at all possible, instead, replacing them with "men" and "women".

We have updated the text to avoid there pronouns in most places.

The article in general would benefit from a disclosure statement early on conceding that gender investigated here is only as a binary variable, discounting its spectrum.

See Line 74-76.

Please also report how gender balance is defined in the datasets as in the data summary in supplementary materials, within the main text.

Our definition of gender balance (proportion of researchers who are men,) is given on Line 103.

(12) The data summary Table S1 could benefit from explaining the variables in the first column. It is currently unclear how granularity, size of dataset and quotas/pre-allocation? are defined.

These lines have been removed as they information they contained is included elsewhere in the table with far better explanations!

(13) There are only 4 data points for investigating covariation between gender balance and funding success in CIHR. This should be discussed as a limitation.

The small size of the dataset is now mentioned on Line 348.

(14) L455 "Research varies widely across disciplines" in terms of what?

This sentence has been extended.

(15) L456 Maybe I am missing something but I don't understand the relevance of "Physicists' search for the grand unified theory" to research quality.

Removed.

(16) Can you provide more discussion into the results of your bibliographic analysis and Figure 2? An explanation into the relationships seen in the figure at least would be helpful.

Thank you we have clarified the relationships seen in each of figures 2A (Lines 226-235), 2B (Lines 236-252), and 2C (lines 260-268).

(17) It would be helpful to include in the discussion a few more sentences outlining:- Potential future research that would help disentangle mechanisms behind the trends you find.- How this research could be applied. Should there be some effort to standardise?

We have added a short paragraph to the discussion about implications/applications, and future research (Lines 481-484).

(18) The introduction could benefit from discussing and explaining their a priori hypotheses for how research from female-biassed disciplines may be evaluated differently.

While not discussed in the introduction, possible explanations for why and how research in female dominated fields might be evaluated differently are explored in some detail in the Discussion. We think once is enough, and towards the end is more effective than at the beginning.

(19) L16 "Our work builds on others' findings that women's work is valued less, regardless of who performs that work." I find this confusing because in your model, there is a significant interaction effect between gender:pmen. This suggests that for female-biassed disciplines, there is even more of a devaluation for women, which I think your lines in figure 1 suggest.

Correct but men are still affected, so the sentence is correct. What is confusing is that the finding is counter to what we might expect.